# *In vivo* expansion of functionally integrated GABAergic interneurons by targeted increase in neural progenitors

Rachel E Shaw[1], Benjamin Kottler[2], Zoe N Ludlow[2], Edgar Buhl[3], Dongwook Kim[2], Sara Morais da Silva[4], Alina Miedzik[1], Antoine Coum[1], James JL Hodge[3], Frank Hirth[2,*] & Rita Sousa-Nunes[1,**]

## Abstract

A central hypothesis for brain evolution is that it might occur via expansion of progenitor cells and subsequent lineage-dependent formation of neural circuits. Here, we report *in vivo* amplification and functional integration of lineage-specific circuitry in *Drosophila*. Levels of the cell fate determinant Prospero were attenuated in specific brain lineages within a range that expanded not only progenitors but also neuronal progeny, without tumor formation. Resulting supernumerary neural stem cells underwent normal functional transitions, progressed through the temporal patterning cascade, and generated progeny with molecular signatures matching source lineages. Fully differentiated supernumerary gamma-amino butyric acid (GABA)-ergic interneurons formed functional connections in the central complex of the adult brain, as revealed by *in vivo* calcium imaging and open-field behavioral analysis. Our results show that quantitative control of a single transcription factor is sufficient to tune neuron numbers and clonal circuitry, and provide molecular insight into a likely mechanism of brain evolution.

**Keywords** circuit plasticity; evolutionary neurobiology; lineage expansion; neural stem cells; Prospero

**Subject Categories** Development & Differentiation; Neuroscience

**The EMBO Journal (2018) 37: e98163**

## Introduction

Insect and vertebrate brains are characterized by a common principle of repetitive, modular organization, clearly visible, for example, in their optic lobes and visual cortex, respectively (Sanes & Zipursky, 2010). These consist of neural circuits arranged in columns formed by repeated divisions of neural stem and progenitor cells and are thus lineage-related (Ito & Awasaki, 2008). Neurons from these ontogenetic clones often share common physiological functions such as feature detection selectivity (Gao *et al*, 2013). It has been proposed that this clonal unit architecture may be key to understand brain evolution (Rakic, 1995). Accordingly, the multiplication of progenitor cells and subsequent lineage-dependent formation of neural circuits are thought to underlie the emergence of complex behavioral repertoires and cognitive capacity (Chittka & Niven, 2009; Grillner & Robertson, 2016).

Stem and progenitor cell numbers are regulated by a variety of mechanisms that can depend on symmetric versus asymmetric modes of division, direct versus indirect neurogenesis, proliferation rates and their time-windows of activity, as well as differentiation and apoptosis (Hirth, 2010; Sousa-Nunes *et al*, 2010; Krubitzer & Dooley, 2013; Strausfeld & Hirth, 2013; Arai & Pierani, 2014; Hoerder-Suabedissen & Molnár, 2015). Neural stem cell (NSC) properties are determined by cell extrinsic and intrinsic mechanisms that pattern and specify them and their behaviors. There are several well-known examples of how varying levels of a single extracellular signaling molecule can lead to different cellular fates and properties during development (Shilo & Barkai, 2017). The effect of different signaling levels and/or of convergence of various signaling pathways is usually described as a combinatorial output of transcription factors on specific enhancers with a binary outcome (ON or OFF) for regulating expression of other factors and determines cell identity (Perrimon *et al*, 2012). The consequence of differing transcription factor levels on progenitor cell properties, and tissue scale has been less well characterized. Neural precursor-specific attenuation of β-catenin expression in mice leads to expansion of the neural progenitor pool, increase in brain surface area without increase in cortical thickness,

1   Centre for Developmental Neurobiology, Institute of Psychiatry, Psychology & Neuroscience, King's College London, London, UK
2   Department of Basic and Clinical Neuroscience, Maurice Wohl Clinical Neuroscience Institute, Institute of Psychiatry, Psychology & Neuroscience, King's College London, London, UK
3   School of Physiology, Pharmacology and Neuroscience, University of Bristol, Bristol, UK
4   Department of Physiology, Development and Neuroscience, University of Cambridge, Cambridge, UK
    *Corresponding author. Tel: +44 2078480789; E-mail: frank.hirth@kcl.ac.uk
    **Corresponding author. Tel: +44 2078486567; E-mail: rita.sousa-nunes@kcl.ac.uk

and consequent generation of gyri and sulci in the normally smooth cortex (Chenn & Walsh, 2002). However, it has not been determined whether progenitor cell expansions *in vivo* result in reiteration of clonal units with functionally integrating neural circuits.

Here, we report for the first time *in vivo* neural progenitor expansion and neural circuit multiplication with lineage resolution analyses. We performed stepwise manipulations of the levels of the homeodomain transcription factor Prospero (Pros), a key player in neuronal specification and a neural tumor suppressor in *Drosophila* (Sousa-Nunes & Hirth, 2016). Pros is expressed in NSCs but asymmetrically segregated into transient progenitors where it drives neuronal differentiation (Li & Vaessin, 2000; Choksi *et al*, 2006). It is well described that Pros mutation or attenuation can lead to failure in transient progenitor differentiation and tumor formation, with supernumerary NSCs formed *at the expense of neurons* (Bello *et al*, 2006; Choksi *et al*, 2006; Bayraktar *et al*, 2010). We show that Pros can be tuned down to levels resulting in expansion not only of progenitors but also of *neurons* and, within a certain limit, do so in the absence of tumorigenesis. In fact, an amount of Pros below *in situ* immunodetection is sufficient for neuronal differentiation. Our work thus demonstrates that high levels of Pros are required not for neuronal differentiation but to preclude transient progenitor reversal into supernumerary NSCs; still higher levels are required to preclude tumorigenesis. Utilizing this tool, we modulated Pros expression in the *engrailed* NSC lineages that generate GABAergic interneurons in the central complex of the adult brain. Expression profiling and functional assays demonstrate that control over Pros levels can be achieved to generate supernumerary progenitors as well as supernumerary neurons that retain molecular and functional properties of the lineages of origin while avoiding tumor formation. Within a non-tumorigenic range of attenuation, Pros titration led to commensurate modulation of the proportion between progenitor and neuron number. We further show that supernumerary GABAergic interneurons constitute functional ring neurons that integrate into ellipsoid body circuitry; and present behavioral analyses that demonstrate efficient sensory-motor transformation and motor action selection by the thus expanded central complex microcircuit.

## Results

### Engrailed protocerebral lineages as a model for targeted NSC expansion *in vivo*

We first aimed to characterize model lineages for our study and to define tools and timings for the desired manipulations. We focused on Engrailed (En) protocerebral lineages because of our knowledge concerning their stem cells of origin, projection pattern, and neurotransmitter identity, plus the ability to manipulate them. In the developing brain of *Drosophila*, the homeodomain transcription factor En is expressed in neural lineages that define the posterior compartment of neuromere boundaries (Hirth *et al*, 1995, 2003). We have found that the posterior protocerebral En lineages contribute to the formation of GABAergic ring (R) neurons of the ellipsoid body of the central complex in the adult brain (preprint: Kottler *et al* 2017). To target these lineages for spatiotemporally controlled genetic perturbations in R neuron progenitors, we employed a

widely used enhancer trap line in which GAL4 activity mimics the expression of *engrailed* (Brand A, communication to Flybase.org).

Lineage tracing showed that the DPLam, DALv, and BAla remain as the only proto- and deutocerebral En-positive clusters from embryo to adult (Fig 1A and B). The embryonic and early larval DALv and BAla clusters contain two NSCs each, distinguishable by larger cell size and expression of NSC markers such as the basic helix–loop–helix transcription factor Deadpan (Dpn) or the adaptor protein Miranda (Mira). The DPLam cluster never includes NSCs as confirmed by lineage tracing (schematized in Fig 1B). In accordance with a previous report (Kumar *et al*, 2009), we found persistent *en > mCD8::GFP* expression in the two DALv NSCs (of the DALv2 and DALv3 lineages) throughout larval stages until the NSCs disappear by a terminal differentiative division at pupal stages (Maurange *et al*, 2008). In contrast, from larval stages onwards *en > mCD8::GFP* expression was undetectable in the two BAla NSCs (of the BAla3 and BAla4 lineages, Pereanu & Hartenstein, 2006), as summarized schematically in Fig 1C. Some cells downregulate *engrailed* during development, accounting for broader reporter expression in permanently labeled lineages than in *en > mCD8::GFP* (Fig 1B versus C).

Lineage analysis together with marker gene expression revealed that DALv2 and DALv3 correspond to Engrailed-expressing neuroblasts Ppd5 and Ppd8 that can be identified by their position and the expression of the *Drosophila* Pax2/5/8 orthologue Pox Neuro (Poxn) in their progeny (Fig 1A; Urbach *et al*, 2003). Ppd5 and Ppd8 can be distinguished by expression of Dachshund (Dac) (Urbach *et al*, 2003), with the Dac-negative Ppd5 NSC leading to progeny that form projections contributing to the formation of the ellipsoid body (preprint: Kottler *et al* 2017; Fig 1A arrow). The ellipsoid body in *Drosophila* is a ring-like neuropil and part of the adult central complex that mediates sensorimotor transformation and the selection and maintenance of behavioral actions (Strausfeld & Hirth, 2013; Fiore *et al*, 2015). The neurons that project to the ellipsoid body, tangential R neurons, are GABAergic interneurons that form subtype and layer-specific synapses along the ring neuropil (preprint: Kottler *et al* 2017). Together these data suggest that *en-Gal4* is a suitable tool to target identified NSC lineages in order to expand *in vivo* the number of Ppd5-derived DALv2 larval lineages and their R neuron progeny.

### Pros downregulation in DALv2/v3 lineages can expand neuronal number

*En*-expressing NSC lineages are of the so-called larval type I, characterized by the expression of Pros and the basic helix–loop–helix transcription factor Asense (Ase; Fig 1D) as well as of a single transient progenitor called ganglion mother cell (GMC; Sousa-Nunes *et al*, 2010). We wondered whether Pros levels could be manipulated to amplify *En*-expressing NSC lineages. In mitotic type I NSCs, Pros is cortically and asymmetrically localized via the adaptor protein Mira, which ensures its nuclear exclusion in self-renewing NSC and its selective segregation into the smaller GMC daughter. In GMCs, Mira is degraded and Pros translocates into the nucleus where it promotes neuronal differentiation (Choksi *et al*, 2006). Pros also translocates into the nucleus of NSCs themselves just prior to their final/terminal differentiative division at pupal stages (Maurange *et al*, 2008). Loss of function of *pros* prevents GMC differentiation with consequent generation of supernumerary NSC-like cells at the expense of

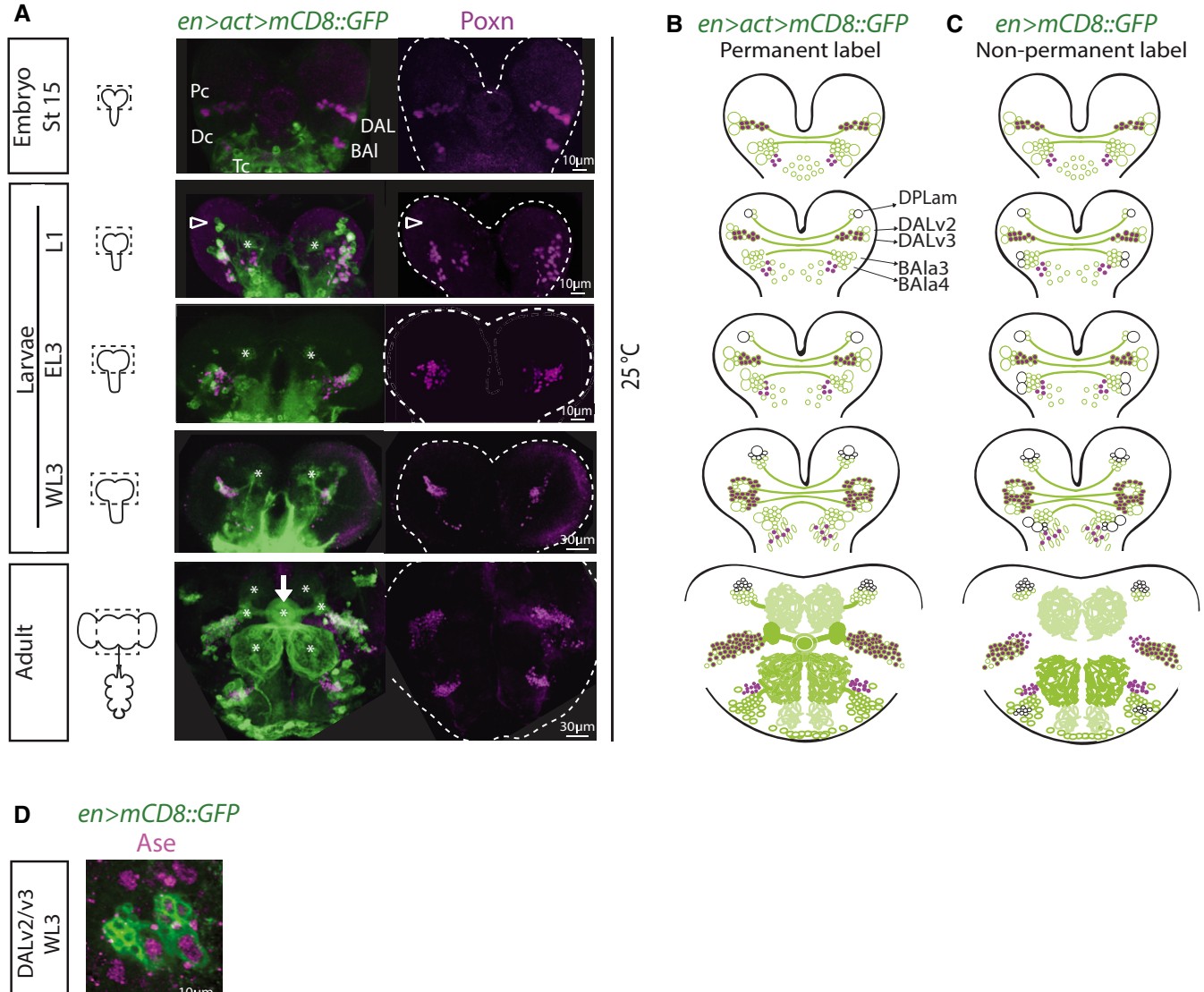

**Figure 1. Engrailed lineages used as model for targeted expansion *in vivo*.**

A   Left: Schematics of CNSs of various stages with dotted outlines indicating central brain regions indicated in (B). Right: Time course pictures of brains containing permanently labeled cells in the En expression domain co-stained for Poxn (maximum intensity projections; split magenta channel). At embryonic stage 15, clusters of *en* cells in the protocerebrum (Pc), deutocerebrum (Dc), and tritocerebrum (Tc) can be detected, which have been named in antero-posterior order: (i) P/PC/b1/DALv (for dorso-antero-lateral)/MC (for medial cluster—because of later emergence of a cluster anterior to this one—see below); (ii) D/DC/b2/BAla (for baso-antero-lateral)/PC (for posterior cluster, a nomenclature which could be confused with that for the protocerebral cluster); and (iii) T/TC/b. In first-instar larvae (L1), an additional protocerebral cell cluster is visible antero-dorsal to the DALv (arrowhead), which starts expressing *en* after embryonic stage 15, and that has been named DPLam (for dorso-postero-lateral)/AC (for anterior cluster). Asterisks, neuropil structures; arrow, ellipsoid body of the central complex (adult structure).

B   Schematics of pictures shown in (A) in which larger circles represent NSCs (green if labeled with GFP reporter, black if not) and smaller circles represent neurons (magenta if Poxn[+]). Only protocerebral and deutocerebral lineages schematized.

C   Schematic representation of expression time course of *en>mCD8::GFP* (therefore, non-permanently labeled lineages) with same coding as described in (B). Only protocerebral and deutocerebral lineages schematized.

D   Picture of DALv2/v3 NSCs (large cells) showing Ase expression, characteristic of type I NSCs.

neurons (Bello *et al*, 2006; Betschinger *et al*, 2006; Choksi *et al*, 2006; Lee *et al*, 2006; Fig EV1A). These supernumerary NSCs can persist aberrantly into adulthood and propagate by transplantation into wild-type (WT) hosts (Fig EV1B and C). Indefinite propagation by serial transplantation, genetic instability, and metastatic potential defines them as tumorigenic (Caussinus & Gonzalez, 2005).

Conversely, overexpression of *pros* in NSCs leads to premature termination of proliferation and results in less progeny (Li & Vaessin, 2000; Bello *et al*, 2006; Choksi *et al*, 2006).

We manipulated *pros* levels by UAS-mediated RNAi (Dietzl *et al*, 2007; Ni *et al*, 2009) and tuned them by various means: (i) employing two RNAi lines of different strengths that

target all *pros* isoforms, one weaker here termed LH, and one stronger here called KK (Fig EV2A–C); (ii) rearing animals at different temperatures to modulate GAL4 efficacy for UAS-mediated RNAi (Wilder, 2000); and (iii) co-expressing or not UAS-Dicer-2 (Dcr-2) to enhance RNAi potency (Dietzl *et al*, 2007). We observed that DALv2/v3 NSCs reactivate from quiescence during the second-instar larval stage (L2). Thus, by growing *en-GAL4* embryos at 18°C, then placing hatched larvae at various temperatures, we found supernumerary NSCs specifically of DALv2/v3 lineages (Fig 2A). Expanded DALv2/v3 lineages fuse, therefore data for both were pooled and they are hereon referred to simply as DAL.

As expected, the stronger the *pros* attenuation, the more the DAL cells labeled by the NSC marker Dpn (Fig 2B). Yet in *prosRNAi*, as in WT lineages, DAL NSCs can be seen to divide asymmetrically (Movies EV1 and EV2). Counterintuitively for the downregulation of a neuronal differentiation factor, within a range, downregulation of Pros led to increased absolute numbers of neurons within the DAL lineage domain. This was evidenced by the absence of Dpn and the expression of Embryonic Lethal Abnormal Vision (Elav; Fig 2B and C), a marker for terminally differentiated neurons. Raising the temperature before L2 led to supernumerary NSCs in BALa3/a4 but never in DPLam (Fig 2D), where *en* is only expressed in neurons. This is in agreement with the previous observation that *pros* downregulation is insufficient to cause neuronal dedifferentiation (Carney *et al*, 2013). Together, these data demonstrate that Pros levels can be tuned for targeted amplification of identified progenitor cells that retain the ability to generate neuronal progeny.

That Pros downregulation results in aberrant proliferation of GMCs is known (see above). What is novel and surprising here is that Pros levels could be lowered to below detectability by immunohistochemistry and still allow neuronal differentiation, even supernumerary neuronal production.

## Supernumerary NSCs generated by Pros downregulation are not necessarily tumorigenic

We next sought to characterize the supernumerary cells generated upon *pros* attenuation. Because Pros can act as a neural tumor suppressor, we first asked whether there was a range of its downregulation within which supernumerary progenitors would be non-tumorigenic. We first assessed whether Pros-attenuated DAL cells incorporated the thymidine analog 5-ethynyl-2′-deoxyuridine (EdU) ectopically at adult stage—a sign of evading termination and thus, a first indication of possible immortality. All WT central brain NSCs terminate divisions at pupal stages (Ito & Awasaki, 2008), upon which Dpn expression is no longer detectable. However, in $en > mCD8::GFP,prosRNAi^{KK}$ animals reared at 29°C, ectopic EdU$^+$ cells were observed in adult brains (Fig 2E). In agreement with the EdU proliferation assay, ectopic Dpn$^+$ cells in the adult brain were also only observed under the stronger *pros* attenuation conditions (Fig 2E). The same was the case for strong *pros* downregulation with the *actin5C* FLP-out system even with the weaker $prosRNAi^{LH}$ if animals were reared at 25°C or above (Fig EV2D). In contrast, like in controls, EdU$^+$ and Dpn$^+$ cells were never seen in $en > mCD8::GFP,prosRNAi^{LH}$ (no FLP-out) adult brains (Fig 2E). This suggested that within levels of *pros* attenuation, supernumerary NSCs are always able to undergo terminal divisions. To determine this directly, we counted DAL NSCs in control $en > mCD8::GFP$ and $en > mCD8::GFP,prosRNAi^{LH}$ animals at various times during pupal stages. In controls, the two DAL NSCs were identified by Dpn expression at onset of puparium formation (white pre-pupal stage, WPP), usually just one NSC remained 24 h after puparium formation (APF) and none beyond 48 h APF (Fig 2F). When $en > prosRNAi^{LH}$ animals were reared at 25°C, around thirty Dpn$^+$ DAL cells were observed at WPP, approximately half at 24 h APF, fewer still by 48 h APF and none by 72 h APF (Fig 2F).

---

**Figure 2. Pros attenuation can increase DAL neuron number and can lead to non-tumorigenic supernumerary DAL NSCs.**

A  Pictures of WL3 stage WT and expanded DAL lineages with *pros* knock-down at different temperatures (single optical sections; lineages outlined with dotted lines) and schematic of expanded ones with same coding as in (B).

B  Quantification of Dpn$^+$ and Dpn$^-$ cells in DAL lineages in the conditions depicted in (A); histograms represent the mean (*n* = 4–12 CNSs; details in source data) and error bars SEM.

C  Pictures of WL3 stage WT versus expanded DAL lineages with *pros* knock-down to levels compatible with the production of numerous supernumerary neurons (single optical sections; lineages outlined with dotted lines in split magenta and blue channels).

D  Pictures of adult WT and *prosRNAi*-expressing DALam lineages in animals reared at 29°C (maximum intensity projections).

E  In adult DAL lineages (outlined with dashed lines), like in control, $prosRNAi^{LH}$ knock-down never leads to EdU- or Dpn-positive cells; in contrast, with $prosRNAi^{KK}$ knock-down a few EdU$^+$ cells can be observed (boxed area shown at higher magnification in inset) as well as numerous Dpn$^+$ cells, some of which aberrantly retain the early temporal marker Chinmo (maximum intensity projections).

F  Quantification of Dpn$^+$ cells in WT and expanded DAL lineages at various timepoints after WPP stage; histograms represent the mean (*n* = 6–34 CNSs; details in source data) and error bars SEM.

G  Schematic representation of expression of key transcription factors (color-coded) in type I neural lineages (NSCs, GMCs, and neurons). Top row: Following asymmetric localization during NSC metaphase, Pros (blue) is segregated exclusively into the GMC daughter, in which Pros translocates into the nucleus promoting a differentiative division to generate two neurons; at pupal stages, preceding NSC terminal division, Pros translocates into the NSC nucleus, which leads to differentiation of the NSC itself and thus absence of NSCs in the adult central brain. Middle row: In conditions of *pros* attenuation (by GAL4-dependent RNAi, which can be exacerbated by increasing the temperature from 18 to 29°C; by overexpression of Dcr-2; and/or filtering *en-GAL4* through the strong *act5C* promoter with the FLP-out method that also results in permanent cell labeling), GMCs can revert back to NSCs (purple square arrow) and the frequency with which this occurs is inversely proportional to Pros levels (black and gray triangles); accompanying GMC reversal is the production of supernumerary NSCs and supernumerary neurons (magenta and blue shapes, respectively) up to the point when absence of Pros is incompatible with neuronal differentiation, at least in some lineages (blue shape tapering off in parallel with Pros levels approximating zero). Bottom row: Illustration of how Pros levels can direct cellular decisions in type I lineages, from WT with a single NSC, to formation of non-tumorigenic supernumerary cells, to tumor formation.

Data information: See main text for other abbreviations and see also Figs EV1 and EV2, and Movies EV1 and EV2.
Source data are available online for this figure.

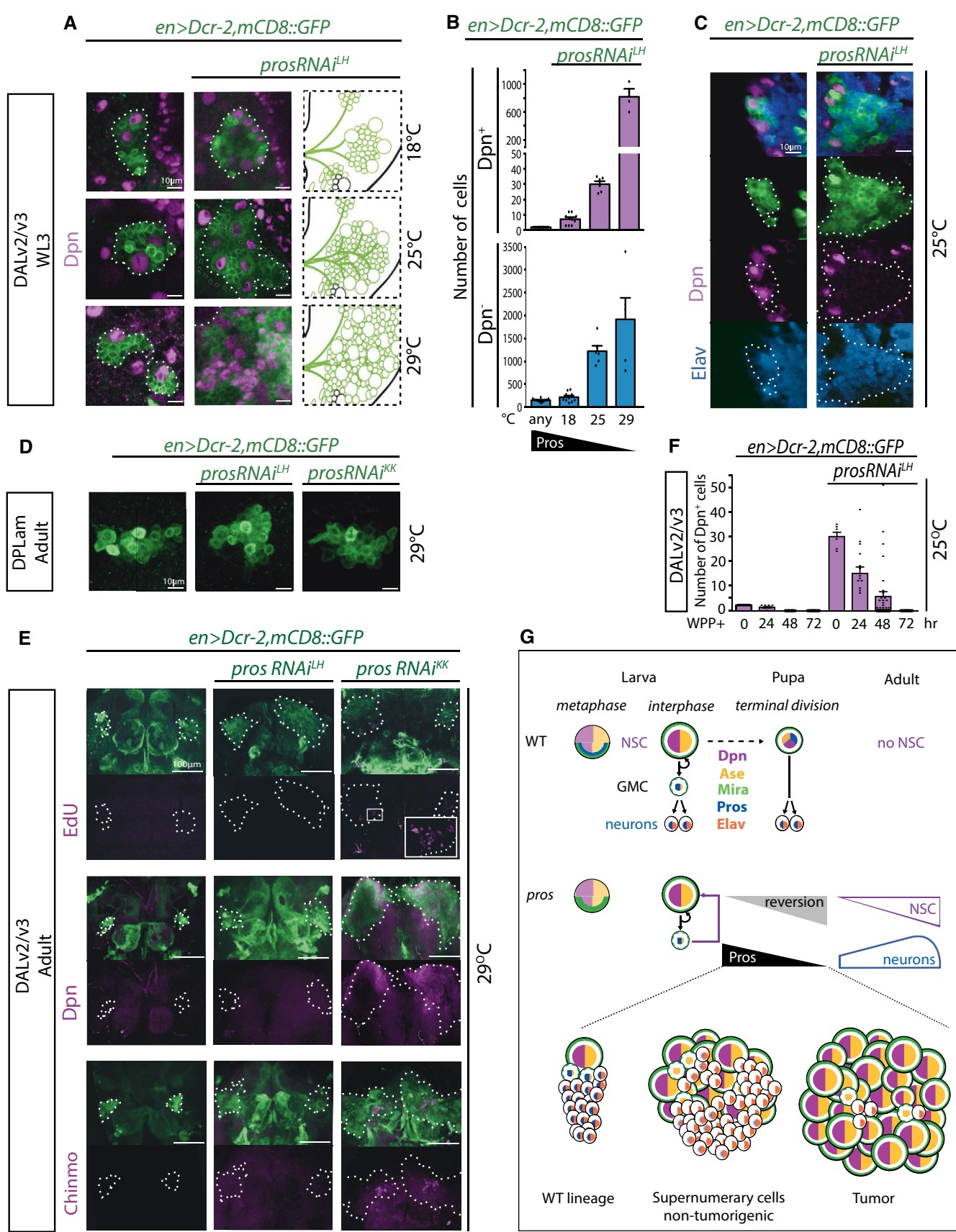

**Figure 2.**

We and others have observed that persistence into adulthood and tumorigenesis of supernumerary NSCs only occurs if they are generated at early larval stages (Fig EV1D–F; Narbonne-Reveau *et al*, 2016). It has been proposed that aberrant maintenance of the early NSC marker chronologically inappropriate morphogenesis (Chinmo) is required for tumorigenesis (Narbonne-Reveau *et al*, 2016). We observed Chinmo expression in a subset of ectopic Dpn$^+$ cells in the adult brain in *pros*-null (*pros$^{17}$*) clones as well as in strong knock-down conditions with *prosRNAi$^{KK}$*, but not in the milder knock-down with *prosRNAi$^{LH}$* (Figs 2E and EV1B). The latter was not because of supernumerary DAL NSCs being exclusively generated at late larval stages when they are non-tumorigenic, as demonstrated by a time course of DAL NSC numbers in *en > mCD8::GFP,prosRNAi$^{LH}$* (Fig EV2E). This indicates that, directly or indirectly, Pros regulates Chinmo expression, with a threshold level of Pros required for Chinmo downregulation in NSCs. Altogether, these data demonstrate that Pros levels can be manipulated to generate supernumerary NSCs that terminally differentiate and are thus non-tumorigenic for the animal (summarized in Fig 2G).

## Supernumerary NSCs generated by Pros downregulation have normal physiological responses

Having established a model of lineage-restricted expansion of non-immortal NSCs, we next determined whether these possessed functional properties of WT counterparts. An important change in *Drosophila* NSC behavior during development is their switch from larval feeding-dependent to feeding-independent proliferation. This underlies brain sparing under nutritional deprivation (Cheng *et al*, 2011), a phenomenon also observed in human fetuses (Cohen *et al*, 2015). *Drosophila* brain sparing occurs from approximately 12 h (at 25°C) after the molt into third-instar larval stage (L3). Larval starvation after this time does not delay development and results in smaller adults that notwithstanding have a normal-sized CNS (Cheng *et al*, 2011). Remarkably, NSCs proliferate at the same rate under these circumstances as in fed animals, which is not the case if animals are starved earlier. We reproduced the observation of NSC feeding-dependent proliferation in L2 animals and feeding-independent proliferation at early L3 (EL3) + 12 h specifically in DAL lineages and found the same behavior in *en > mCD8::GFP,prosRNAi$^{LH}$* expanded DAL lineages (Fig 3A). These data indicate that supernumerary DAL NSC lineages feature physiological properties similar to non-expanded, WT NSCs.

## Non-tumorigenic supernumerary NSCs generated by Pros downregulation progress normally through lineage-specific temporal pattern

Termination of divisions at pupal stage and transition of supernumerary NSCs into larval feeding-independent proliferation suggest that they mature temporally. *Drosophila* NSCs employ so-called temporal series that consist of sequential expression of transcription factors that impart identity to the neurons produced at the time of their expression in progenitors and that also impact NSC properties (Maurange *et al*, 2008; Tsuji *et al*, 2008; Kohwi & Doe, 2013). Termination requires progression through the temporal series components Castor (Cas) and Seven-up (Svp) (Maurange *et al*, 2008), which are also required for progression from the early marker Chinmo to later markers. We therefore reasoned that non-tumorigenic supernumerary NSCs likely progressed through the temporal series, which would be in contrast to tumorigenic *pros* loss-of-function NSCs that aberrantly retain Chinmo expression. Aberrant temporal patterning of NSCs would be predicted to result in aberrant identity of their neuronal progeny and thus not lead to *in vivo* expansion of defined neuronal types.

To directly test temporal progression in expanded DAL lineages, we needed to first characterize their temporal markers. We performed a time course expression analysis of a number of transcription factors previously implicated in NSC temporal patterning or known to be expressed in the postembryonic *Drosophila* CNS (Maurange *et al*, 2008; Bayraktar & Doe, 2013; Li *et al*, 2013). The summary of expression time courses is schematized in Fig 3C. Svp had the briefest pulse of expression in DAL NSCs, centered around the L2/L3 molt (Figs 3B and EV3). Eyeless (Ey) and Grainyhead (Grh) were detected in NSCs at all timepoints examined (Fig EV3); on the other hand, Eyes Absent (Eya) and the glial marker Reversed Polarity (Repo) were never detected (R.E.S unpublished observations). Poxn, Twin-of-Eyeless (Toy), Chinmo, Dichaete (D), Dac, Svp, Extradenticle (ExD), and Broad-Complex (Br-C) displayed temporally restricted expression; Poxn was only detected at EL3 (Fig EV3). In agreement with what has been described for ventral nerve cord NSCs (Maurange *et al*, 2008), Chinmo and Br-C were largely mutually exclusive; Chinmo was detected at EL3, and the transition to Br-C took place at mid-L3 (ML3; defined as EL3 + 24 h at 25°C) with Br-C remaining detectable, albeit weakly, at 10 h APF (Fig 3C and D). Expression of D and Toy resembled that of Chinmo, and all three were undetectable from wandering third-instar larval stage (WL3), i.e., late larval stage, onwards. Dac (in the one DAL NSC) and ExD were detected as early as Chinmo, D, and Toy; however, their expression was prolonged beyond puparium formation.

---

**Figure 3. Non-tumorigenic supernumerary DAL NSCs display temporal properties of original NSCs.**

A   Volume at WL3 stage (fold-change relative to mean value of WT lineages in animals fed from late L2 stage: first histogram) of DAL lineages of animals that were split at the L2/L3 molt into late L2 or EL3 and then either fed (F) or starved (S) thereon; histograms represent the mean (*n* = 7–16 CNSs; details in source data) and error bars SEM. Red bars indicate mean statistically significantly different from control; ***$P$ < 0.001 unpaired *t*-test.

B   Percentage of Svp$^+$ cells in WT and expanded DAL lineages at timepoints centered on the L2/L3 molt (10 h before, at the molt, and 10 h after); histograms represent the mean (*n* = 10 CNSs; details in source data) and error bars SEM.

C   Schematic representation of expression time course in WT and expanded DAL lineages of a number of transcription factors in DAL lineage NSCs; hatched color represents lower levels than full color.

D   Pictures of expression time course in WT and expanded DAL lineages of the temporal markers Chinmo and Br-C (single optical sections; lineages outlined with dotted lines in split green, magenta and blue channels); co-stained for Mira allows identification of the NSCs.

Data information: See main text for abbreviations and see also Fig EV3.
Source data are available online for this figure.

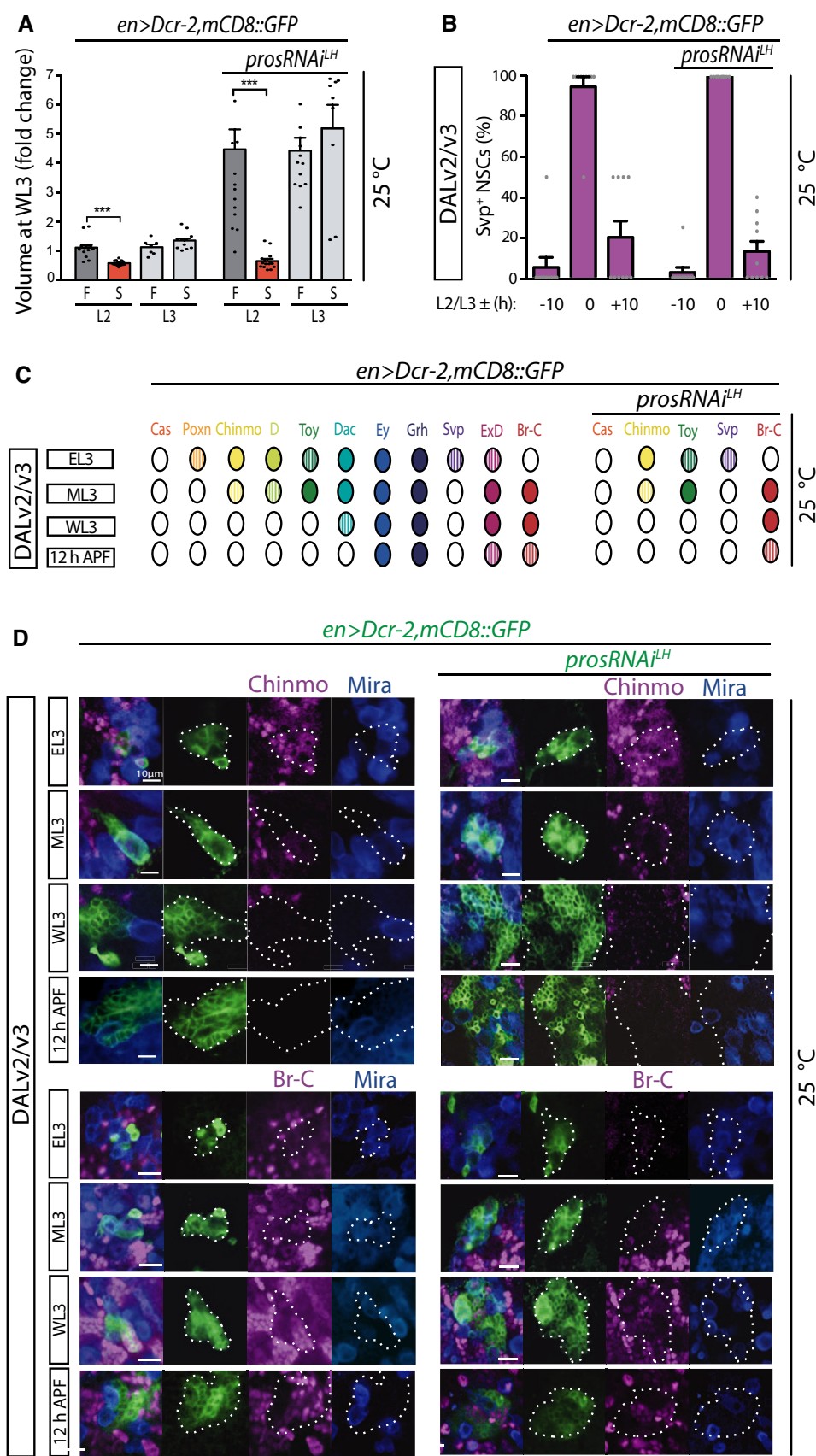

**Figure 3.**

The overall pattern of expression of the above markers was analogous between DALv2 and DALv3 NSCs, as seen by expression in both at the examined timepoints, albeit not absolutely synchronous, occasionally switching on or off in one within a few hours of the other. Cas was not detected within this time-window (Fig EV3), suggesting that if transiently expressed in these cells, it would likely be prior to EL3, consistent with what has been observed in ventral nerve cord NSCs (Maurange *et al*, 2008).

Having laid out a WT temporal pattern for larval DAL NSCs, we next scored non-tumorigenic supernumerary DAL NSCs generated by *pros* downregulation to determine whether it was analogous. In contrast to strong *pros* attenuation to levels resulting in NSC termination failure, non-tumorigenic supernumerary DAL NSCs never retained Chinmo expression at late larval/early pupal stages (Figs 2E and G, and 3C). This is also in agreement with what we described above concerning Chinmo absence at adult stages in non-tumorigenic *pros* attenuation. Toy and Br-C expression in supernumerary NSCs also followed that of WT and even the timing of the short Svp pulse was recapitulated (Figs 3B and EV3). Also like in WT DAL NSCs, Cas, Eya, and Repo were never observed in supernumerary DAL NSCs during this time-window (R.E.S unpublished observations). Altogether, we conclude that temporal patterning of non-tumorigenic Pros-attenuated supernumerary DAL NSCs proceeds like that of their corresponding WT counterparts.

## Supernumerary DAL NSCs generate increased numbers of lineage-specific GABAergic interneurons

Temporal patterning of progenitors defines the identity of their neuronal progeny. We therefore predicted that supernumerary DAL NSCs should generate supernumerary Poxn and GABA-expressing ellipsoid body R neurons (preprint: Kottler *et al* 2017), which upon differentiation would be able to project into the ellipsoid body ring neuropil.

As predicted, the number of DAL cells expressing Poxn and GABA in the adult brain increased with *pros* attenuation in a dose-dependent manner (Fig 4A and B, Movies EV1 and EV2). Furthermore, most FLP-out events (required to visualize the ellipsoid body neuropil; Fig 4C), accompanying Pros attenuation to non-tumorigenic levels, labeled neurite tracts that match with those of WT, projecting medially via the bulb and terminating in the characteristic ring neuropil of the ellipsoid body (Fig 4B, Movies EV1 and EV2). The supernumerary projections in the *pros*-attenuated genotype form a thicker neuropil than in WT that then converges onto the ellipsoid body. Occasionally we also observed ectopic neuropil (Fig EV4) and a ventrally open, arch-like ellipsoid body (Fig 4B open arrow), like that seen in most arthropods which have been proposed to be the ancestral condition (Strausfeld, 2012). These data show that targeted amplification of Ppd5-derived DAL NSCs can lead to increased numbers of Poxn- and GABA-expressing neurons characterized by axonal projections and terminal arborization patterns like those of GABAergic R neurons of the ellipsoid body in the adult brain of *Drosophila*.

## Supernumerary R neurons are physiologically active

We next determined whether supernumerary GABA-expressing interneurons were physiologically active within the expanded ellipsoid body circuitry. To this end, we carried out functional imaging using the fluorescent protein calcium sensor GCaMP6f (Chen *et al*, 2013; Figs 4D and E, and EV5, Movies EV3 and EV4). A characteristic feature of GABAergic R neurons is their inhibition by GABA-A receptor signaling (preprint: Kottler *et al* 2017). To investigate whether the supernumerary interneurons were physiologically active, and thus responded to GABAergic inhibition, we applied picrotoxin, a non-competitive blocker of GABA-A receptor chloride channels.

For both control and experimental genotypes, picrotoxin application resulted in increased GCaMP6f fluorescence in cell bodies (Figs 4D and E, and EV5, Movies EV5 and EV6) and the expanded ring neuropil (E.B. unpublished observations). Activity occurred rapidly across the population, with GCaMP6f fluorescence intensity increasing significantly above baseline for cell bodies upon picrotoxin administration (Figs 4D and E, and EV5, Movies EV5 and EV6). The pattern and time course of this calcium increase were comparable between control and lineage-expanded brains at the population and single-cell levels (Figs 4D and E, and EV5). Regardless of their position within the GCaMP6f-labeled pool of cells, all recorded neurons showed robust response to picrotoxin (Figs 4E and EV5, single-cell traces). Together, these data suggest that downregulation of Pros in Ppd5-derived DAL NSCs can be tuned to expression levels that result in non-tumorigenic supernumerary progenitors which in turn generate increased numbers of lineage-specific GABAergic interneurons that are physiologically active.

## Supernumerary R neurons are well tolerated for efficient sensory-motor integration and motor action selection

Finally, to gain insights into the behavioral impact of integrated supernumerary R neurons, we analyzed the motor behavior of lineage-expanded flies. Ellipsoid body R neurons have been implicated in goal-directed behavior, higher motor control, spatial orientation learning and memory, as well as in attention, arousal, and decision-making, suggesting that R neurons are involved in the assessment of sensory information for adaptive behavior (Strausfeld & Hirth, 2013; Fiore *et al*, 2015). Indeed, previous studies showed that R neuron dysfunction affects sensory-motor transformation and higher motor control (Fiore *et al*, 2015; preprint: Kottler *et al* 2017).

$en > Dcr-2, prosRNAi^{LH}$ flies showed impaired motor behavior and tremor-like motions in comparison to controls (Z.N.L. unpublished observations). This could be due to widespread expression of the *en-GAL4* driver that targets various tissues beyond the CNS, including in muscle progenitors, which also express Pros, or within posterior CNS lineages. To further confine Pros attenuation, we restricted GAL4 activity to the head by employing *teashirt (tsh)-GAL80* (Clyne & Miesenbock, 2008; Fig EV6) and video-tracked open-field behavior of freely moving animals using the Drosophila Arousal Tracking (DART) system (Faville *et al*, 2015). While this assay can report differences in motor behavior upon EB circuit perturbations (preprint: Kottler *et al* 2017), analysis of controls and animals with supernumerary R neurons revealed no significant differences (Fig 5A–D). Together with the observed functional anatomy (Fig 4), these data suggest that Pros-attenuated, supernumerary GABAergic interneurons are physiologically active and integrate into ellipsoid body R neuron circuitry without detriment to sensory-motor integration and higher motor control.

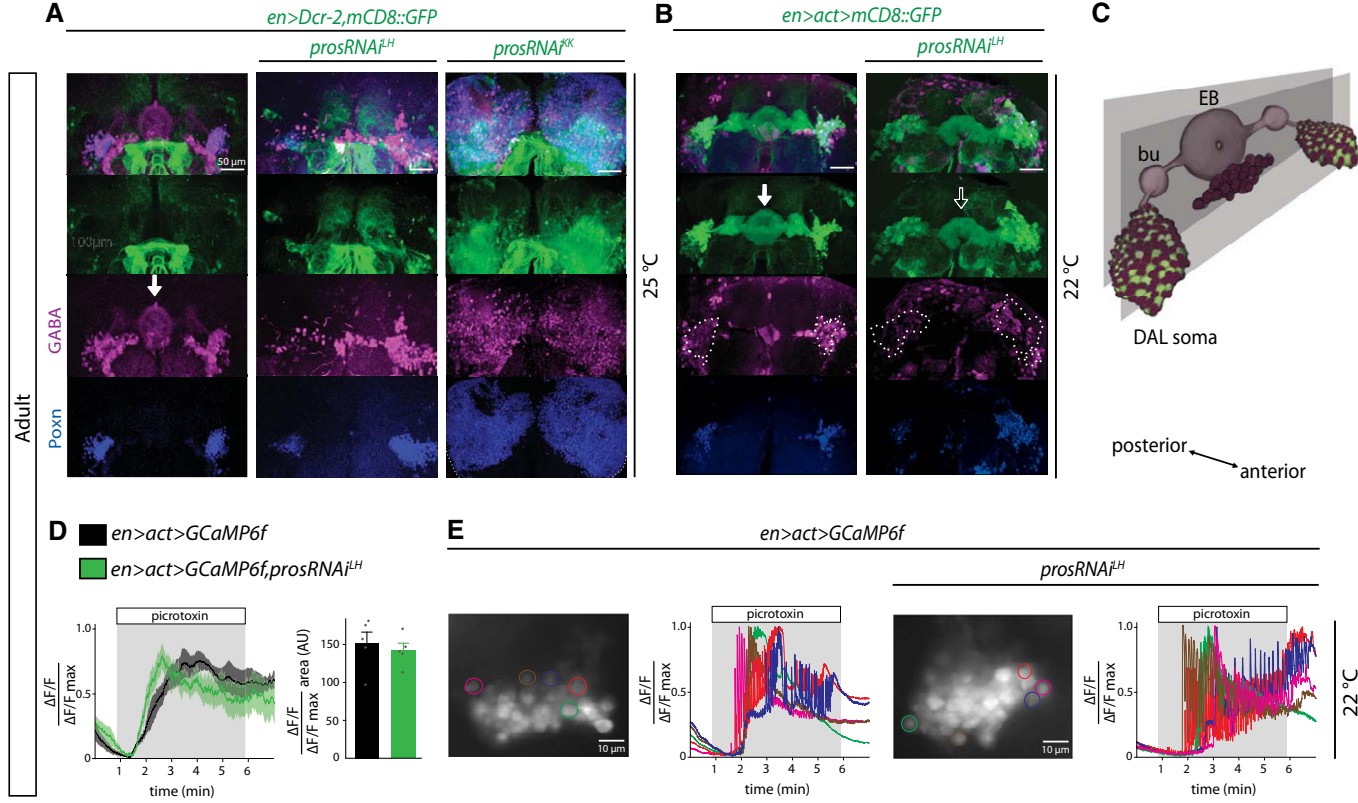

**Figure 4. Supernumerary DAL neurons are functional.**

A, B   Pictures of adult WT and expanded DAL lineages showing expression of GABA in a subset of Poxn[+] cells (maximum intensity projections of a few optical sections; split green, magenta, and blue channels). (A) Lineages expanded with the *en-GAL4* driver alone, in which GFP does not label the EB, although thus structure is discernible by GABA staining (arrow). (B) Lineages expanded and permanently labeled with *en>act>GAL4*, in which the EB is clearly visible (arrows). Note that there are GABA-expressing cells outside the DAL domain (which is outlined by dotted lines in the magenta channel for clarity).

C   Schematic representation of the EB and bulb (bu), which are neuropil structures deriving from projecting GABAergic DAL neurons (magenta spheres). Note that the DAL cell bodies and the EB are located in different antero-posterior planes and that there are GABA-expressing cell bodies outside the DAL domain. Also, most of the cell body expansion occurs in an antero-posterior direction (see Movies EV3 and EV4) so volume difference is not fully captured in maximum intensity projections.

D   Relative intensity of GCaMP6f expression in adult DAL lineages for control (black, solid line is mean, and shaded area is SEM) and experimental (green) genotypes over 7 min of recording, including response to picrotoxin (added at 1 min and washed out at 6 min: gray background), each normalized to their own maximum intensity. Histograms represent the mean integration of the area under the curves (*n* = 5 CNSs; details in source data) during picrotoxin application and error bars SEM (no statistical difference; unpaired two-tailed *t*-test: *P* = 0.6068).

E   Maximum intensity projections of GCaMP6f expression in adult DAL lineages of WT and expanded DAL lineages with *pros* knock-down, with color outline of five example cells per genotype whose individual normalized calcium traces are shown (details in source data).

Data information: See also Figs EV4 and EV5, and Movies EV3–EV6.
Source data are available online for this figure.

# Discussion

Our results show that reduction in Pros levels in Ppd5-derived larval lineages can result in non-tumorigenic supernumerary NSCs and derived progeny that differentiate into Poxn and GABA-expressing R neurons able to functionally integrate into the ellipsoid body ring neuropil of the adult central complex in *Drosophila*.

## Pros levels direct cellular decisions

It has been proposed that nuclear Pros levels distinguish three NSC behaviors: absent for self-renewal, low for quiescence, and high for differentiation (Lai & Doe, 2014). Pros is undetectable and has no

known function in proliferating NSC nuclei; its deletion does not convert type I into type II NSC lineages comprising transit-amplifying cells in addition to GMCs, nor does it cause neuronal dedifferentiation (Bowman *et al*, 2008; Carney *et al*, 2013). Our data therefore suggest that the *in vivo* lineage expansion described here is regulated by Pros levels in GMCs. Each GMC must undertake a number of binary decisions: to undergo a differentiative division into two neurons/glia or to revert back to a NSC state; if reverting to a NSC, whether to go quiescent or divide; if dividing, whether to give rise to two NSC-like cells or to a NSC plus a GMC; to progress temporally or not.

Our results indicate that some GMCs reverted back to NSCs but that others were able to perform differentiative divisions. The proportion of these decisions depended on Pros levels as seen by

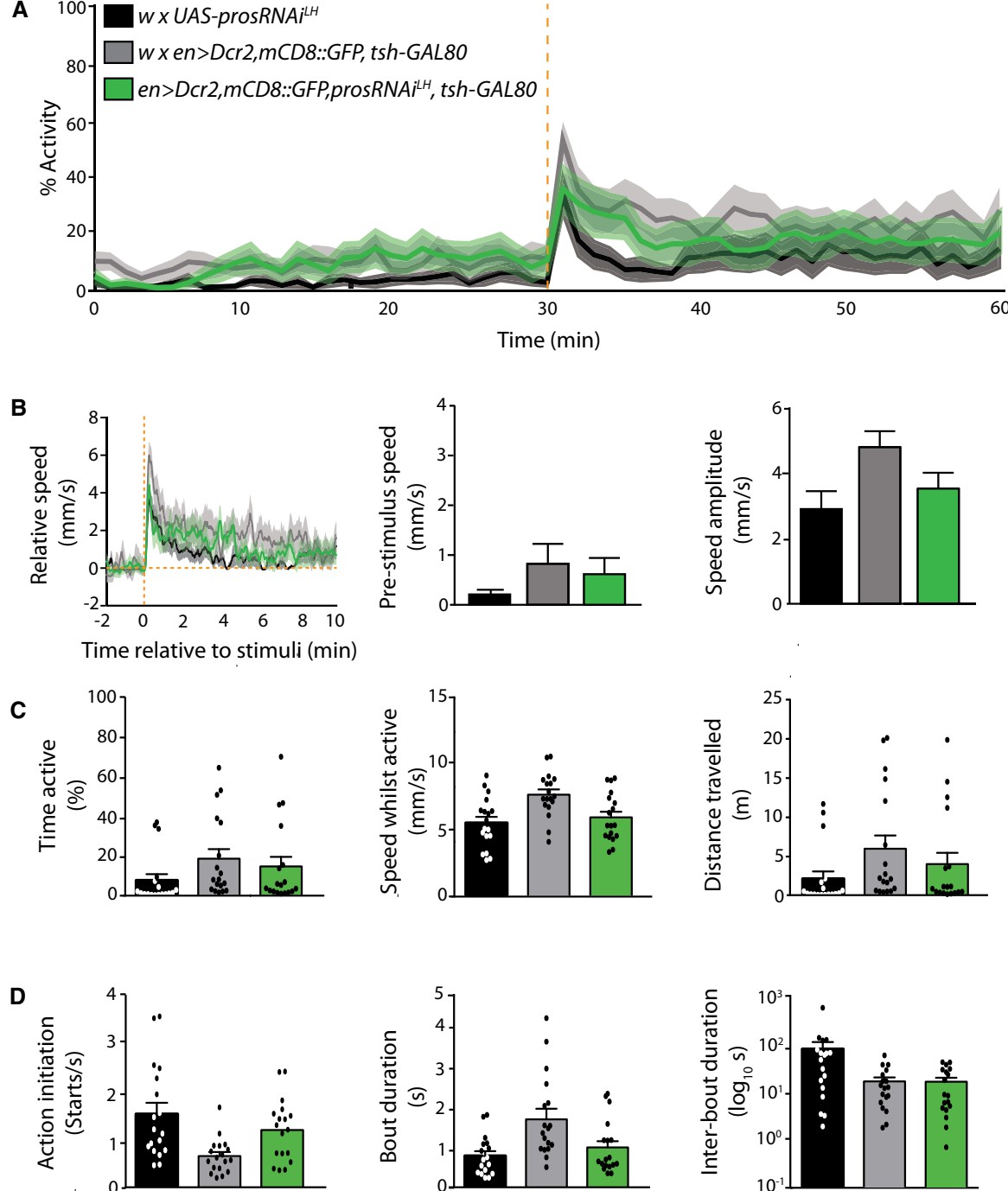

**Figure 5. Non-tumorigenic DAL lineage expansion is well tolerated for behavioral output of central complex circuitry.**

A   Adult activity over 60 min of recording, including response to mechanical stimulation after 30 min (dashed orange line), from which the nine parameters below were extracted (*n* = 18 individuals per genotype, combined from two independent experiments with nine individuals per genotype per experiment, which led to the same overall result; details in source data).

B   Stimuli-related measurements. Left: relative speed before and after the stimulus; center: pre-stimuli speed; right: amplitude of the response. Error bars SEM.

C   Overall activity measurements. Left: overall activity duration; center: speed while active; right: total distance travelled; histograms represent the mean and error bars SEM.

D   Individual action measurements. Left: frequency of starts; center: duration of an action; right: duration of pauses in between actions; histograms represent the mean and error bars SEM.

Data information: See also Fig EV5. Values for experimental genotype (green) were not statistically significantly different from controls (black and gray; Kruskal–Wallis test). *n* = 18.

Source data are available online for this figure.

the graded response observed with RNAi strength, temperature, or absence versus presence of Dcr-2. However, we cannot make the correspondence between that proportion and quantity or concentration of nuclear Pros, as above 25°C this was undetectable by immunohistochemistry. This in itself indicates that only a small amount of Pros is required for neuronal differentiation and that the role of high levels of Pros is to preclude GMC reversion and to ensure Chinmo downregulation. Varying Pros levels in GMCs can therefore result in markedly distinct cell population outcomes, from single (in WT) to multiple lineages that are temporally patterned like WT, to a block in temporal progression, neuron abolishment and tumorigenesis. Pros levels can make the difference between Chinmo-mediated malignancy (with aberrant neuronal fates) and non-malignant neural expansion (with compliance of neuronal temporal fates). This is an important finding as the consequence for cell fate is dramatic, and distinct from the finding that we and others made about the differential effect of timing of Prospero attenuation (this work and Narbonne-Reveau *et al*, 2016).

Given the considerable variability in cell and nuclear size in *pros* loss-of-function lineages, it will be interesting to determine whether it is a threshold effect defined by total number of molecules or the concentration of Pros per nucleus what directs these cellular decisions.

### *In vivo* lineage expansion as a tool

This is the first report of *in vivo* neural expansion with lineage resolution. Our findings reveal a Pros dose response for progenitor cell progression, and consistent with previous reports (Narbonne-Reveau *et al*, 2016), they demonstrate that even complete deletion of Pros can be tumorigenic or not, depending on the time of ablation. While our study exploits lineage expansion in an invertebrate, it is reasonable to envisage a similar rationale in the mammalian nervous system. Similar to Pros, mouse Prox1 also regulates the balance between neural progenitor cell self-renewal and neuronal differentiation (Kaltezioti *et al*, 2010). However, irrespectively of whether or not via Prox1, it seems likely that manipulation of this balance could lead to lineage amplification in vertebrates also. In mammals, cell-cycle manipulations have been employed for progenitor expansion (Artegiani *et al* 2011; Nonaka-Kinoshita *et al*, 2013). As exemplified here, manipulation of neuronal differentiation and asymmetric NSC division components is worth investigating with the purpose of expanding progenitors and neurons, and whether lost lineages could be replaced like by like. Indeed, pioneering work on *Drosophila* NSCs led to the identification of conserved molecules and mechanisms also implicated in asymmetric divisions of mammalian NSCs (reviewed in Sousa-Nunes & Hirth, 2016). Given the high degree of evolutionary conservation among the mechanisms that control progenitor cell proliferation and lineage specification in insects and mammals (Sousa-Nunes & Hirth, 2016), we hypothesize that the concept presented here is likely general and valid in vertebrates.

It will also be interesting to determine whether neural progenitor expansion in the adult brain is tractable in a similar way than during development, which is of particular interest for regenerative purposes including of targeting neurodegenerative diseases. In mammals, adult neurogenesis is confined to very few brain regions that produce neurons able to integrate into defined pre-existing networks (van Praag *et al*, 2002; Toni & Schinder, 2015). Our results

provide a rationale to investigate whether and how adult mammalian neural progenitors can be safely expanded *in vivo* to generate neurons of defined identity, and whether these are capable of integrating into functional circuits to restore damaged or lost function.

### Cloned neurons can contribute to behaviorally relevant circuitry

Our proof-of-principle study demonstrates *in vivo* lineage expansion as a means to generate more neurons of defined identity that can integrate into neural circuitry. We show that supernumerary GABAergic ring neurons are physiologically active and integrate into the ellipsoid body circuit without affecting motor behavior even when the animal is exposed to sensory stimulation like mechanical shock (Fig 5). It is unlikely that the addition of supernumerary cells over one generation would improve a circuitry that evolved over millions of years. What was surprising was the observed developmental plasticity and functional robustness of the circuit over a range of cell numbers. This demonstrates that the nervous system of *Drosophila* can show considerable hysteresis in tolerating substantial changes in neuron number while maintaining network properties and functional output. A possible implication of this work is that the neural circuits studied may not wire together solely by deterministic recognition cues, but may be influenced by other currently unknown factors that might be even stochastic in nature (Hassan & Hiesinger, 2015). We believe our results can, however, be rationalized by the anatomy of the ellipsoid body in the central complex of the adult brain. R neuron subtypes form concentric synaptic layers that constitute the ring neuropil which can be subdivided into wedges that correspond to segments of sensory space (Strausfeld, 2012; Seelig & Jayaraman, 2015). Our findings reveal that Pros attenuation results in an enlarged diameter of the ring neuropil (Movie EV3 versus EV4), suggesting that supernumerary R neuron projections add to the layered circuitry without affecting its physiological function. If layer-specific markers were available, it would be interesting to determine how supernumerary R neurons contribute to the individual layers, R1-R4, and thus acquire a subtype-specific identity, or whether they formed new, additional layers.

Several developmental and genetic mechanisms have been proposed for neural circuit evolution. These include inter-progenitor pool wiring whereby a fraction of neurons derived from one progenitor pool migrate away and integrate into a remote brain domain to establish new neuronal wiring (Suzuki & Sato, 2017). In our study, however, supernumerary ring neurons remain at their site of origin and send projections to the EB ring. These observations suggest another mechanism, namely duplication of an entire circuit module (Tosches, 2017). Lineage-related R neurons constitute layers of the ellipsoid body circuitry and thus can be regarded as ontogenetic clones that form a circuit module of the adult brain. The fact that the EB circuit can accommodate a range of cell numbers reveals a plasticity that might have promoted (and carry on doing so) evolutionary adaptation. In fact, the similarity in temporal marker expression between the DALv2 and v3 lineages lends itself to the hypothesis the two might have originated by duplication of an ancestral lineage that subsequently diversified projections and acquired different functions. Indeed, the primary tracts of DALv2 and DALv3 are juxtaposed before DALv3 bifurcates into so-called *supra-* and the sub-ellipsoid secondary axon tracts (Lovick *et al*, 2013). Such multiplication and functional reuse of an existing

feature is a known process in evolution, called exaptation (Gould & Vrba, 1982). It has been suggested that whenever circuit duplication is followed by exaptation, the properties of the circuit would initially remain unaltered (Tosches, 2017). In accordance with this hypothesis, we do not observe gross alterations for the supernumerary ring neurons in their transition through the temporal cascade and the resulting molecular signature, such as GABA and Poxn expression, nor do we observe gross changes in their physiological properties or behavioral readout with the assay applied. It is therefore tempting to speculate that amplification of ontogenetic clones such as lineage-related ring neurons, followed by exaptation of the resulting circuits could be an adaptive mechanism underlying brain and behavioral evolution (Strausfeld & Hirth, 2013; Grillner & Robertson, 2016).

# Materials and Methods

### *Drosophila melanogaster* strains

Unless otherwise stated, all strains were obtained from the Transgenic RNAi Project (TRiP) at Harvard Medical School, Vienna *Drosophila* Resource Centre (VDRC) and Bloomington stock centers. The following strains were used *en-GAL4$^{e16e}$* (gift from A. Brand); *UAS-mCD8::GFP*; *UAS-prosRNAi$^{KK}$* (VDRC 101477); *UAS-prosRNAi$^{LH}$* (TRiP Long hairpin); *UAS-Dcr2* (Bloomington 24646); *y,w,hs-FLP$^{1.22}$;tub-GAL4,UAS-NLS::GFP::6xmyc;FRT82B,tubP-GAL80$^{LL3}$/(TM6B)* (gift from G. Struhl); *pros$^{17}$* (gift from C. Doe) as well as *mira2L44* and *miraYY227* (gifts from W. Chia) recombined with FRT82B; *FRT82B, pros$^{1L32}$* (this study), *UAS-FLP* (Bloomington 8208); *UAS-GCaMP6f* (Bloomington 42747); *act5c > CD2 > GAL4* (*Bloomington 4780*); *tsh-GAL80* (gift from J. Simpson).

### Rearing and staging of *Drosophila*

To assist larval genotyping, lethal chromosomes were re-established over balancer chromosomes marked by *Dfd-YFP*. For larval staging experiments, crosses were performed in cages with grape juice plates (25 % (v/v) grape juice, 1.25 % (w/v) sucrose, 2.5 % (w/v) agar) supplemented with live yeast paste. Embryos were raised at 18°C; larvae hatched within 10 h at 18°C were transferred to our standard cornmeal food (8 % (w/v) glucose, 2 % (w/v) cornmeal, 5 % (w/v) baker's yeast, 0.8 % (w/v) agar in water) and placed at the desired experimental temperature. Starvation consisted of larval rearing in 1 % (w/v) Agar in PBS. For some experiments, development was further synchronized by selecting early L2 (EL2) or EL3 animals morphologically from mixed L1/L2 or L2/L3 molting populations, respectively. Starvation versus feeding experiments was performed on animals selected at the L2/L3 molt, whereby L2 was either fed or starved and EL3 was fed for 12 h before then being starved or remaining fed. ML3 was defined as EL2 + 24 h at 25°C (or equivalent at other temperatures, with larval development taking twice as long at 18°C than at 25°C) and WL3 by wandering behavior of late L3 (which corresponds roughly to EL2 + 48 h at 25°C (or equivalent at other temperatures). Data from males or females of the same genotype were pooled without distinction.

### Lineage tracing

*en-GAL4* was combined with *UAS-FLP* and a FLP-out cassette (*act > STOP > GAL4*) under control of the strong *actin5C* promoter to permanently label cells that at any point expressed *en*, i.e., *en > act > mCD8::GFP*.

### Identification of *pros$^{1L32}$* allele

*FRT82B,pros$^{1L32}$* was recovered from an unpublished ethyl methanesulfonate forward genetic screen (RSN, W.G. Somers and W. Chia). It was recognized as a candidate *pros* allele by the supernumerary NSC phenotype followed by complementation testing with *pros$^{17}$*. The identity of the genetic lesion was determined for this study by sequencing of genomic DNA of heterozygous *FRT82B/FRT82B,pros$^{1L32}$*.

### Generation of *pros* and *mira* clones

CNS clones were generated by the Mosaic Analysis with a Repressible Cell Marker (MARCM) technique. Virgin females of the MARCM stock *y,w,hs-FLP$^{1.22}$;tub-GAL4,UAS-NLS::GFP::6xmyc;FRT82B,tubP-GAL80$^{LL3}$/(TM6B)* were crossed to *FRT82B* recombined (experimental) or *FRT82B* (control) males and left to lay at 25°C. Larval progeny reared also at 25°C were heat-shocked once in a 37°C water bath for 20 min, either at EL2 or at ML3 and dissected either at WL3 or at adult (schematized in Fig EV1D).

### EdU labeling and immunohistochemistry

For EdU experiments, dissected CNSs were incubated for 2 h in 10 mM EdU/PBS, followed by fixation and color reaction according to the manufacturer's instructions (Click-iT EdU Imaging Kit, Invitrogen). For combined EdU and immunohistochemistry, antibody stains were performed before EdU color reaction. Larval and adult CNSs were fixed in 4 % formaldehyde in PBS for 15 and 25 min, respectively; washes of larval and adult CNSs were performed in 0.1 % or 0.5 % Triton in PBS (PBT), respectively, and antibodies were diluted in 5 % normal goat serum in PBT. Rabbit polyclonal anti-peptide antibodies against *Drosophila* Mira were generated using peptides QRLRFRPTPSHTDTAT (aa99–114) and CVPSPPQKQVLKARNI (aa816–830) of the protein (UniProtKB/Swiss-Prot O45116), with KHL as carrier (Eurogentec, Seraing/Belgium); these were used at 1/200. Guinea pig polyclonal antibodies against *Drosophila* Dpn were generated using DNA encoding amino acids 78–431 cloned into pET29a (Novagen) for protein expression and purification generated by J. Skeath; we either utilized antibodies kindly provided by J. Skeath or ones generated for our laboratory by Dundee Cell Product; these were used at 1/1,000 and 1/5,000, respectively. Other primary antibodies used were as follows: mouse anti-Br-Core (25E9.D7), mouse anti-Dac, mouse anti-Exd (EXDHDcc), mouse anti-Ey, mouse anti-Eya (eya10H6), mouse anti-Pros 1/5 (mAbMR1A), mouse anti-Repo 1/20 (8D12), mouse anti-Svp 1/10 (2D3), all from Developmental Studies Hybridoma Bank (DSHB), Iowa University; guinea pig anti-Ase 1/2,000 (gift from J. Knoblich); mouse anti-Mira 1/50 (mAb81; gift from F. Matsuzaki); rabbit anti-Poxn 1/2,000 (F. Hirth, published); mouse anti-Poxn 1/100 (Hassanzadeh *et al* 1998); rabbit anti-Grh 1/200 (F. Hirth, published); rabbit anti-GABA 1/1,000 (Sigma-Aldrich); rabbit anti-Chinmo 1/1,000 (gift from T. Lee); rat

anti-Chinmo 1/500 (gift from N. Sokol); rabbit anti-Dichaete 1/200 (Nambu and Nambu, 1996); guinea pig anti-Toy (gift from U. Walldorf) 1/1,000. Secondary antibodies used were goat anti-mouse, anti-rabbit, or anti-rat, conjugated with 488, 568, or 647 Alexa fluorochromes at 1/400 (Invitrogen) or 1/200 (Jackson Immuno-Research). CNSs were mounted in Vectashield (Vector Laboratories).

### Image acquisition and analyses

Fluorescence samples were scanned and recorded with Leica TCS SP or Zeiss LSM scanning confocal microscopes. Optical section steps ranged from 0.5 to 2 μm, recorded in line average mode with picture size of 1,024 × 1,024 pixels. Captured images from optical sections were arranged and processed (adjustment of brightness and contrast and occasional despeckle function) using Fiji or ImageJ software; clone volumes were determined with AMIRA software. Figures were arranged and labeled using Adobe Illustrator, Photoshop CS5, and PowerPoint.

### Cell number and clone volume quantifications

Cells were counted using the ImageJ plug-in "Cell Counter". Due to variable size of $Dpn^+$ cells (diameter between ~4 and ~10 μm), they were counted manually by marking and tracking each cell per section in a z-stack. $Dpn^-$ cells had a consistent size (diameter ~3 μm) and so were represented on average in three optical sections taken with 1 μm steps; therefore, all $Dpn^-$ cells were manually marked in each slice, and the total added and divided by 3. Clone volumes were determined with Amira software.

### Allograft assays

A glass Pasteur pipette was pulled to a diameter of 0.5 mm. WL3 containing either control, *pros* or *mira* MARCM clones were washed in PBS, and their CNSs dissected and transferred also into PBS. Before transplantation, young female adult hosts were etherized. One piece of cut VNC per adult host was injected tangentially in the mid-ventral abdomen. After recovering from anesthesia, the hosts were kept under standard *Drosophila* culture conditions at 25°C.

### Calcium imaging

Flies (*en > act > GCaMP6f* and *en > act > GCaMP6f prosRNAi^LH*) were reared at 22°C and imaged at 5–10 days post-eclosion. They were then decapitated and the brains dissected in saline solution containing (in mM): 101 NaCl, 1 $CaCl_2$, 4 $MgCl_2$, 3 KCl, 5 glucose, 1.25 $NaH_2PO_4$, 20.7 $NaHCO_3$ (pH adjusted to 7.2). Brains were placed ventral side up in the recording chamber, secured with a custom-made anchor, and continuously perfused. Picrotoxin (500 μM in saline) was bath-applied through the perfusion system. The calcium fluorescence signal was acquired using an optiMOS camera (QImaging) and a 470 nm LED light source (Colibri, Zeiss) on an upright Zeiss Examiner microscope with a ×63 lens and recorded with MicroManager (8 frames/s, 50–100 ms exposure). Responses to picrotoxin of individual neurons or of the whole DAL population were analyzed using ImageJ, and the fluorescent signal ($\Delta F/F$) was normalized by the maximum $\Delta F/F$ for that brain; 50–59 cells per

brain were analyzed for the control genotype and 53–74 for *prosRNAi* brains. For quantification and comparison between control and experimental responses, the area under the normalized curve was calculated for the whole DAL population for the 5-min period where picrotoxin was applied. All chemicals were purchased from Sigma, and the experiments were performed at room temperature (20–22°C).

### Open-field behavioral analysis

Following eclosion, adult mated females were reared at 18°C for up to 3 days in a 12-h/12-h light cycle. Flies were gassed and separated at least 24 h prior to the experiment and transferred individually into a vial. Separate experiments were performed at the same time of day. Sixty minutes prior to the experiment, flies were transferred to custom-made acetal copolymer (POM-C) open-field arenas (Tecaform AH) to acclimatize. Arenas were 3.5 mm in diameter and 1.5 mm height covered with a transparent acrylic sheet containing 1 mm holes for aeration, assembled into a platform of 36 (arenas). The platform was placed on a white light plate that provided uniform cold light illumination within a temperature-controlled incubator (Stuart Scientific). Video-assisted motion tracking, and analyses were carried out using the DART system amended with custom-made MATLAB (MathWorks) scripts. Video recordings were carried out using a Logitech c920 camera at 10 frames/s for 1 h. The positions of flies were extracted every two recorded frames (0.20 s). A mechanical stimulation was delivered after 30-min recording, composed of a train of five stimulations at 3 V for 200 ms each, spaced by 800 ms.

### Kinematic calculations

To determine behavioral parameters, we calculated fly motion as follows. The inter-frame displacement, $D_i$ (which is calculated between the $i^{th}$ and $(i-1)^{th}$ video frames), was calculated using:

$$D_i = \sqrt{(X_i - X_{i-1})^2 + (Y_i - Y_{i-1})^2} \qquad (1)$$

where $X_j/Y_j$ denote the X/Y coordinates (respectively) for frame *j*. The inter-frame speed was calculated using equation (1) as follows:

$$V_i = \frac{D_i}{(T_i - T_{i-1})} \qquad (2)$$

where $T_j$ denotes the time stamp for frame *j*. A fly was considered "active" for a given video frame if equation (2) exceeded 2 mm/s, equivalent to traveling a body length every second. Absolute speed was calculated by averaging equation (2) for each video frame over all flies/genotype. *Relative speed* is calculated from the absolute speed considering the average pre-stimulus speed as 0 mm/s. *Pre-stimulus speed* represents the average speed of all flies/genotype 2 min prior to mechanical stimulation (calculated using equation 2). *Speed amplitude* was calculated from the absolute post-stimuli speed trace (see next) as the difference between the maximum stimuli response and the average pre-stimuli speed. The absolute

post-stimuli speed was calculated using equation (2), but smoothed (so as to remove high-frequency noise) using the following equation:

$$\hat{V}_i \approx \frac{\left(\sum_{j=0}^{N_{Avg}-1} V_{i-j}\right)}{N_{Avg}} \tag{3}$$

where $N_{Avg}$ is the number preceding video frames ($N_{Avg} = 10$) interpolated to the nearest second. The relative post-stimuli signal was calculated as the absolute post-stimuli speed minus the average pre-stimuli speed. This was fit with a single-inactivation exponential equation with the following functional form:

$$V(\tau) = \left(1 - e^{-\tau/\tau_A}\right)\left(A_0 + A_1 e^{-\tau/\tau_{I1}}\right)H(\tau) \tag{4}$$

where $H(t)$ is the Heaviside step function (=1 if $t > 0$, otherwise 0), $A_{0-2}$ are scale factors, $\tau_A$ is the activation time constant, and $\tau_A/\tau_B$ are inactivation time constants. The exponential equation parameters (the scale factors, time constants, and δt) for (4) were fitted using the MATLAB optimization function, *fit* (MathWorks). These values are not discrete data points but rather values extracted from the average signal fitted to the exponential decay equation (4) with a confidence interval of 95% fitted. *Time active* is defined as the sum of the duration for which the fly is active. *Speed while active* is the average speed during active periods. *Distance travelled* was calculated by summing equation (1) over all active frames for each fly (for the 60 min of recording) and then averaged over all recorded flies/genotype. *Action initiation* was defined as the number of times a fly transitioned from an inactive to an active state divided by the total time spent inactive. *Bout duration* was defined as the length of time an individual fly was continuously active. *Interbout interval* was the time between two bouts of activity during which the fly was continuously inactive (the minimum was 0.2 s, which corresponds to a single video frame).

### Quantifications and statistical analyses

No randomization nor blinding was used. Each data point corresponds to a different individual of the designated genotype or condition. Sample number was arbitrary. Histogram heights correspond to the mean value. With the exception of Fig EV1E (where it would be confusing), sample number is visible by plotting of the data points (those of Fig EV1E are indicated numerically and referred to in the legend). No data points were excluded. In all cases, error bars represent standard error of the mean (SEM), as indicated in figure legends. Data were checked for normalcy. When normal, significance was tested through an unpaired student *t*-test; when not, either with Mann–Whitney or Kruskal–Wallis tests (which is indicated in figure legends). All statistical tests and graphs were generated using Prism software.

**Expanded View** for this article is available online.

### Acknowledgements

We are grateful to A. Barrios, L.Y. Cheng, J. Clarke, C. Desplan, V. Harten-stein, C. Houart, T. Southall, and G. Technau for helpful discussions. We also thank A. Brand, A. Gould, F. Matsuzaki, V. Rodrigues, C. Doe, A. Ghysen, C.Y. Lee, J. Skeath, N. Sokol, and G. Struhl for flies or antibodies, and the TRiP (NIH/NIGMS R01-GM084947), VDRC, and Bloomington Stock Centres for providing fly stocks. This work was supported by grants from the Wellcome Trust Grant 107414/Z/15/Z (to S.M.d.S.); the UK Biotechnology and Biological Sciences Research Council Grant BB/J017221/1 (to J.J.L.H.); UK Medical Research Council (G0701498; MR/L010666/1), the UK Biotechnology and Biological Sciences Research Council (BB/N001230/1), the Royal Society (Hirth2007/R2) (to F.H.); Cancer Research UK Career Development Fellowship and Royal Society (RG2016/R1) (to R.S.N.).

### Author contributions

RES, BK, ZNL, EB, AC, AM, DK, and SMdS performed experiments. RES, BK, ZNL, EB, AC, AM, DK, SMdS, JJLH, FH, and RS-N analyzed data. RS-N and FH supervised the project and wrote the manuscript, to which all authors contributed.

### Conflict of interest

B.K. is co-founder of Burczyk/Faville/Kottler LTD. All remaining authors declare no competing financial interests.

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
