## [Review Process File · The EMBO Journal]

***In vivo* expansion of functionally integrated GABAergic interneurons by targeted increase in neural progenitors**

Rachel. E. Shaw, Benjamin Kottler, Zoe N. Ludlow, Edgar Buhl, Dongwook Kim, Sara Morais da Silva, Alina Miedzik, Antoine Coum, James J. L. Hodge, Frank Hirth and Rita Sousa-Nunes

Review timeline:

Submission date: 5th September 2017
Editorial Decision: 2nd October 2017
Revision received: 21st January 2018
Editorial Decision: 15th February 2018
Revision received: 14th March 2018
Accepted: 28th March 2018

Editor:

Transaction Report:

1st Editorial Decision

2nd October 2017

Thank you for submitting your manuscript to The EMBO Journal. Your study has now been seen by two referees and their comments are provided below.

As you can see from the comments, both referees find the analysis interesting and suitable for publication here. They raise a number of different issues that I am presume you should be able to sort out in a good way. Given the referees' positive recommendations, I would like to invite you to submit a revised version of the manuscript, addressing the comments raised in full. I should add that it is EMBO Journal policy to allow only a single major round of revision and that it is therefore important to address the raised concerns at this stage.

REFeree REPORTS

Referee #1:

In this work Shaw and colleagues report the surprising and very interesting observation that the levels of the transcriptional modulator Prospero dictate neural stem cell behavior in *Drosophila*. They find that attenuating Pros levels with a specific RNAi-mediated manipulation can produce supernumerary neural stem cells, as expected, which however proceed to differentiate in a lineage appropriate fashion to give rise to supernumerary neurons. These neurons appear to integrate and function within the circuit indistinguishably from their wild type sisters. These findings are of great interest to the field as they suggest that different levels of Prospero regulate different aspects of neuronal lineage progression. They also make available a tool to potentially easily manipulate neuronal lineage size independently of neuronal subtype and fate acquisition. Finally, these data suggest one way by which expansion and reduction in neuronal lineages might occur during development and evolution.

Overall, the conclusions are of broad interest to the field and are appropriate for publication in the EMBO Journal. There are however a number of technical and textual concerns that need to be addressed to strengthen the major conclusions.

Major concerns:

The most significant concern is the almost sole reliance on a single RNAi line for most of the findings (Pros-RNAiLH). This line is said to reduce levels of Pros less than the other RNAi line used at the beginning of the study (Pros-RNAiKK). However the images of figure EV2 suggest - if anything - the opposite. More convincing data for the difference in down regulation need to be presented. Alternatively, the data should be supported by showing "LH" behaves more like "KK" or a classic Pros mutant in a Pros heterozygous background. In other words, the authors need to nail down the argument that this really is due to changes in Pros levels, especially as they cannot make direct correlations between Pros levels and single stem cell clone size. Perhaps also showing that Pros KK behaves more like Pros LH at 18 degrees would further support their argument. I may have missed this experiment, but I could not find such data in the manuscript. While I am inclined to agree with the interpretation of the authors, it is best to err on the side of caution and be as sure as possible, since this is the major finding of this work.

Minor concerns:

- Figure 3D: the authors state that Chinmo is expressed with the proper temporal order in "LH" knock-down cells. However the image shows persistence of Chinmo at 12h APF in LH conditions, but not wild type conditions.
- Figures 4D and 5: while the measurements show that the additional neurons have calcium transients similar to wild type cells and - interestingly enough - do not hamper locomotor activity, these are fairly crude measures of neuronal function and the statements about that in the results and discussion section need to be toned down accordingly.
- The fact that extra GABA-ergic neurons have no dramatic effects on circuit function has important implications for neuronal circuit wiring: it strongly suggests a degree of developmental plasticity that can tolerate significant changes in neuronal number with minor effects on circuit wiring. This means wiring specificity in neuronal circuits cannot possibly be solely mediated by deterministic recognition cues. The authors may wish to discuss these broader implications of their findings.

Referee #2:

Shaw et al. studied how Pros regulates certain lineages in the Drosophila brain and suggest that this process might cause differences in GABAergic neurogenesis in different species. By manipulating Pros levels with different RNAi lines and temperatures, the authors found that reducing Pros levels led to expansion of both progenitors and neurons in the lineages. They further suggest that the non-tumorigenic neurons resulted from the supernumerary NSCs were physiologically normal and did not affect sensory-motor integration and motor action selection, indicating that these neurons were functionally integrate into neural circuits.

While the overall concept is interesting, there are several missing links in the story.

First, there is basically no evidence that corroborates the authors' claim that the two prosRNAi lines (KK and LH) reduce Pro to different levels. Since this assumption is prerequisite for a major point of the paper, the authors should show -- either in vivo or in vitro -- Pros knocking down efficiency of the different RNAi lines.

Second, the authors show in figure 5 that supernumerary R neurons did not affect motor behavior, thus claiming that expanded GABAergic neuronal progeny is functional. To validate the behavior quantification, the authors should add a control experiment that demonstrates their experimental system is able to detect changes in these behaviors. For example, they may use flies with dysfunctional R neurons or apply optogenetics.

Third, while the authors speculate that Pros-regulated expansion of neurogenesis might be a mechanism for expanding neural circuits in evolution, alternative possibilities should be discussed.

Although the expanded number of GABAergic neurons may be functional as an ensemble, some of these neurons might not be functional. It is unclear to this reviewer how many neurons were quantified and how they were selected. It seems that there are only 5 samples in Figures 4D and 4E. The quantification and statistical analysis for data shown in these figures need to be strengthened.

"In vivo expansion of functionally integrated GABAergic interneurons by targeted increase of neural progenitors"

[Paper #EMBO]-2017-98163]

We thank both referees for their comments and helpful suggestions how to improve our manuscript. We have addressed all of the points raised and outline our response in detail below.

Referee #1:

In this work Shaw and colleagues report the surprising and very interesting observation that the levels of the transcriptional modulator Prospero dictate neural stem cell behavior in Drosophila. They find that attenuating Pros levels with a specific RNAi-mediated manipulation can produce supernumerary neural stem cells, as expected, which however proceed to differentiate in a lineage appropriate fashion to give rise to supernumerary neurons. These neurons appear to integrate and function within the circuit indistinguishably from their wild type sisters. These findings are of great interest to the field as they suggest that different levels of Prospero regulate different aspects of neuronal lineage progression. They also make available a tool to potentially easily manipulate neuronal lineage size independently of neuronal subtype and fate acquisition. Finally, these data suggest one way by which expansion and reduction in neuronal lineages might occur during development and evolution.

Overall, the conclusions are of broad interest to the field and are appropriate for publication in the EMBO Journal. There are however a number of technical and textual concerns that need to be addressed to strengthen the major conclusions.

Major concerns:

The most significant concern is the almost sole reliance on a single RNAi line for most of the findings (Pros-RNAiLH). This line is said to reduce levels of Pros less than the other RNAi line used at the beginning of the study (Pros-RNAiKK). However the images of figure EV2 suggest - if anything - the opposite. More convincing data for the difference in down regulation need to be presented. Alternatively, the data should be supported by showing "LH" behaves more like "KK" or a classic Pros mutant in a Pros heterozygous background. In other words, the authors need to nail down the argument that this really is due to changes in Pros levels, especially as they cannot make direct correlations between Pros levels and single stem cell clone size. Perhaps also showing that Pros KK behaves more like Pros LH at 18 degrees would further support their argument. I may have missed this experiment, but I could not find such data in the manuscript. While I am inclined to agree with the interpretation of the authors, it is best to err on the side of caution and be as sure as possible, since this is the major finding of this work.

Reply: We appreciate the reviewer's point that RNAi always raises concerns of specificity and requires validation. The two *UAS-RNAi* lines utilized in this work are well validated in that they a) abolish detectable Pros antigen when induced at temperatures optimal for GAL4 activity; b) phenocopy *pros* loss-of-function mutations. They are thus widely accepted in the field and were also employed in the cited Narbonne-Reveau *et al.* 2016 study. Concerning, the differential strength of the 2 lines, we observed differential Pros levels within the targeted lineages (regions outlined by dotted line in EV2B) between *prosRNAi,LH* and *prosRNAi,KK* if animals were reared at lower temperatures (≤ 22 °C) such that anti-Pros staining was detectable in RNAi animals. Stainings were of course carried out in parallel for this comparison and imaged under the same conditions. The dark patches devoid of signal in the LH panel are where NSCs (Dpn⁺ cells) reside in the section; note that **in the Dpn⁻ cells**, anti-pros signal is generally stronger in the LH panel than in the KK. This observation is in good agreement with our functional data to which we have now added the suggested *in vivo*

experiment: showing that *prosRNAi, KK* at lower temperatures behaves more like *prosRNAi, LH* at higher temperatures: whereas *en>Dcr2, CD8::GFP* driving *prosRNAi, KK* at 29 °C led to a number of DAL cells that was impossible to count, this genotype raised at 25 °C or 22 °C led to accountable cell numbers, in a proportion of Dpn+/Dpn- (average of 485/1174 or 19/502, respectively) comparable to that of *prosRNAi, LH* at 29 °C or 25 °C (average of 819/1737 or 30/1222, respectively). This data has now been added as panel EV2C (to be compared with Fig. 2B).

Notwithstanding, we also made every effort to add *in vitro* data to this, by driving both RNAis with the pan-neural driver *elav-GAL4* and quantifying relative Pros levels by Western Blot analysis. In agreement with the *KK* line being stronger than the *LH*, *elav>Dcr2, prosRNAi, KK* brains were visibly enlarged compared with *elav>Dcr2, prosRNAi, LH* or *elav>Dcr2, cherryRNAi* controls. Normalization with a variety of usual loading control genes proved unsuitable. For example, probing the same blot simultaneously for Actin and Tubulin showed disparate proportions of these across genotypes, indicating that at least one if not both were unsuitable for normalization. Indeed, "normalization" attempts with a variety of proteins led to increased ratio of Pros in RNAi samples relative to WT, whereas we know there is less Pros per cell in RNAi genotypes (very clear by immunohistochemistry using the same anti-Pros antibody). We can only speculate about the reason for this. *elav>Dcr2, prosRNAi* broadly induces a fate transformation such that the WT low-NSC/high-neuron numbers ratio is reversed (high-NSC/low-neuron numbers) and if neurons and NSCs express different amounts of the genes used as loading controls they are unsuitable as controls.

Minor concerns:

-Figure 3D: the authors state that Chinmo is expressed with the proper temporal order in "LH" knock-down cells. However the image shows persistence of Chinmo at 12h APF in LH conditions, but not wild type conditions.

Reply: The image showed Chinmo expression in **neurons** (Mira-negative cells) at 12 h APF in LH conditions. Indeed, in both WT and expanded lineages Chinmo persists in early-born **neurons** but not in late **NSCs**. The image chosen for LH happened to show Chinmo+ neurons whereas the WT did not, which may have created confusion so we have replaced the LH image with one where deeper/early cells are not in the plane of view.

-Figures 4D and 5: while the measurements show that the additional neurons have calcium transients similar to wild type cells and - interestingly enough - do not hamper locomotor activity, these are fairly crude measures of neuronal function and the statements about that in the results and discussion section need to be toned down accordingly.

Reply: We have toned down the results and discussion section. The relevant results section is now subtitled "**Supernumerary R neurons are physiologically active**" and we changed the end of the section as follows: "Regardless of their position within the GCaMP6f-labelled pool of cells, all recorded neurons showed robust response to picrotoxin (**Figure 4E single cell traces**). Together these data suggest that downregulation of Pros in Ppd5-derived DAL NSCs can be tuned to expression levels that result in non-tumorigenic supernumerary progenitors which in turn generate increased numbers of lineage-specific GABAergic interneurons that are physiologically active." We also modified the discussion section entitled "Cloned neurons..."; the first two sentences now read: " Our proof-of-principle study demonstrates *in vivo* lineage expansion as a means to generate more neurons of defined identity that can integrate into neural circuitry. We show that supernumerary GABAergic ring neurons are physiologically

active and integrate into the ellipsoid body circuit without affecting motor behavior even when the animal is exposed to sensory stimulation like mechanical shock (**Figure 5**).

-The fact that extra GABA-ergic neurons have no dramatic effects on circuit function has important implications for neuronal circuit wiring: it strongly suggests a degree of developmental plasticity that can tolerate significant changes in neuronal number with minor effects on circuit wiring. This means wiring specificity in neuronal circuits cannot possibly be solely mediated by deterministic recognition cues. The authors may wish to discuss these broader implications of their findings.

Reply: This is an interesting possibility and we now added to the discussion: “This demonstrates that the nervous system of *Drosophila* can show considerable hysteresis in tolerating substantial changes in neuron number whilst maintaining network properties and functional output. A possible implication of this work is that the neural circuits studied may not wire together solely by deterministic recognition cues, but may be influenced by other currently unknown factors that might be even stochastic in nature (Hassan and Hiesinger, 2015). We believe our results can however be rationalized by...”

Referee #2:

Shaw et al. studied how Pros regulates certain lineages in the Drosophila brain and suggest that this process might cause differences in GABAergic neurogenesis in different species. By manipulating Pros levels with different RNAi lines and temperatures, the authors found that reducing Pros levels led to expansion of both progenitors and neurons in the lineages. They further suggest that the non-tumorigenic neurons resulted from the supernumerary NSCs were physiologically normal and did not affect sensory-motor integration and motor action selection, indicating that these neurons were functionally integrate into neural circuits.

While the overall concept is interesting, there are several missing links in the story.

*First, there is basically no evidence that corroborates the authors' claim that the two *prosRNAi* lines (KK and LH) reduce Pros to different levels. Since this assumption is prerequisite for a major point of the paper, the authors should show -- either *in vivo* or *in vitro* -- Pros knocking down efficiency of the different RNAi lines.*

Reply: We appreciate the reviewer's point that RNAi always raises concerns of specificity and requires validation. The two *UAS-RNAi* lines utilized in this work are well validated in that they a) abolish detectable Pros antigen when induced at temperatures optimal for GAL4 activity; b) phenocopy *pros* loss-of-function mutations. They are thus widely accepted in the field and were also employed in the cited Narbonne-Reveau *et al.* 2016 study. Concerning, the differential strength of the 2 lines, we observed differential Pros levels within the targeted lineages (regions outlined by dotted line in EV2B) between *prosRNAi,LH* and *prosRNAi,KK* if animals were reared at lower temperatures (≤ 22 °C) such that anti-Pros staining was detectable in RNAi animals. Stainings were of course carried out in parallel for this comparison and imaged under the same conditions. The dark patches devoid of signal in the *LH* panel are where NSCs (Dpn⁺ cells) reside in the section; note that **in the Dpn⁻ cells**, anti-pros signal is generally stronger in the *LH* panel than in the *KK*. This observation is in good agreement with our functional data to which we have now added the suggested *in vivo* experiment: showing that *prosRNAi,KK* at lower temperatures behaves more like *prosRNAi,LH* at higher temperatures: whereas *en>Dcr2,CD8::GFP* driving *prosRNAi,KK* at 29 °C led to a number of DAL cells that was impossible to count, this genotype raised at 25 °C or 22 °C led to accountable cell numbers, in a proportion of Dpn⁺/Dpn⁻ (average of 485/1174 or 19/502, respectively) comparable to that of *prosRNAi,LH* at 29 °C or 25 °C (average of 819/1737 or 30/1222, respectively). This data has now been added as panel EV2C (to be compared with Fig. 2B).

Notwithstanding, we also made every effort to add *in vitro* data to this, by driving both RNAis with the pan-neural driver *elav-GAL4* and quantifying relative Pros levels by Western Blot analysis. In agreement with the *KK* line being stronger than the *LH*, *elav>Dcr2,prosRNAi,KK* brains were visibly enlarged compared with *elav>Dcr2,prosRNAi,LH* or *elav>Dcr2,cherryRNAi* controls. Normalization with a variety of usual loading control genes proved unsuitable. For example, probing the same blot simultaneously for Actin and Tubulin showed disparate proportions of these across genotypes, indicating that at least one if not both were unsuitable for normalization. Indeed, "normalization" attempts with a variety of proteins led to increased ratio of Pros in RNAi samples relative to WT, whereas we know there is less Pros per cell in RNAi genotypes (very clear by immunohistochemistry using the same anti-Pros antibody). We can only speculate about the reason for this. *elav>Dcr2,prosRNAi* broadly induces a fate transformation such that the WT low-NSC/high-neuron numbers ratio is reversed (high-NSC/low-neuron numbers) and if neurons and NSCs express different amounts of the genes used as loading controls they are unsuitable as controls.

Second, the authors show in figure 5 that supernumerary R neurons did not affect motor behavior, thus claiming that expanded GABAergic neuronal progeny is functional. To validate the behavior quantification, the authors should add a control experiment that demonstrates their experimental system is able to detect changes in these behaviors. For example, they may use flies with dysfunctional R neurons or apply optogenetics.

Reply: In consideration to this point, we silenced these neurons (driver: *en>Dcr2,CD8::GFP;tsh-GAL80*) by expressing a dominant-negative version of *Drosophila* Dynamin (*UAS-shi^{DN}*) or Tetanus-Toxin-Light-Chain (TNT), encoding an inhibitor of synaptic transmission (*UAS-TNT*). Unfortunately no adults eclosed so we were unable to perform behavioral experiments on them. However, we have indeed been able to detect significant behavioral changes upon EB circuit perturbation. Below is an example consisting of inactivation of the GABA-A receptor in EB ring neurons, which will be published elsewhere. We have now added a sentence in the results section (p.18) to this effect: “Whilst this assay can report differences in motor behavior upon EB circuit perturbations (<http://dx.doi.org/10.1101/100420>), analysis of controls and animals with supernumerary R neurons revealed no significant differences (Figure 5).”

Figure for referees not shown.

Third, while the authors speculate that Pros-regulated expansion of neurogenesis might be a mechanism for expanding neural circuits in evolution, alternative possibilities should be discussed.

Reply: We have now discussed alternative possibilities for expanding neural circuits in evolution by expanding the discussion section headed “**Cloned neurons can contribute to behaviorally relevant circuitry**” to include the following: "Several developmental and genetic mechanisms have been proposed for neural circuit evolution. These include inter-progenitor pool wiring whereby a fraction of neurons derived from one progenitor pool migrate away and integrate into a remote brain domain to establish new neuronal wiring (Suzuki and Sato, 2017). In our study, however, supernumerary ring neurons remain at their site of origin and send projections to the EB ring. These observations suggest another mechanism, namely duplication of an entire circuit module (Tosches, 2017). Lineage-related R neurons constitute layers of the ellipsoid body circuitry and thus can be regarded as ontogenetic clones that form a circuit module of the adult brain. The fact that the EB circuit can accommodate a range of cell numbers reveals a plasticity that might have promoted (and carry on doing so) evolutionary adaptation. In fact, the similarity in temporal marker expression between the DALv2 and v3 lineages lends itself to the hypothesis the two might have originated by duplication of an ancestral lineage that subsequently diversified projections and acquired different functions. Indeed, the primary tracts of DALv2 and DALv3 are juxtaposed before DALv3 bifurcates into so-called supra- and the sub-ellipsoid secondary axon tracts (Lovick et al., 2013). Such multiplication and functional reuse of an existing feature is a known process in evolution, called exaptation (Gould and Vrba, 1982). It has been suggested that whenever circuit duplication is followed by exaptation, the properties of the circuit would initially remain unaltered (Tosches, 2017). In accordance with this hypothesis, we do not observe gross alterations for the supernumerary ring neurons in their transition through the temporal cascade and the resulting molecular signature, such as GABA and Poxn expression; nor do we observe gross changes in their physiological properties or behavioral readout with the assay applied. It is therefore tempting to speculate that amplification of ontogenetic clones such as lineage-related ring neurons, followed by exaptation of the resulting circuits could be an adaptive mechanism underlying brain and behavioral evolution (Strausfeld and Hirth, 2013; Grillner and Robertson, 2016).

Although the expanded number of GABAergic neurons may be functional as an ensemble, some of these neurons might not be functional. It is unclear to this reviewer how many neurons were quantified and how they were selected. It seems that there are only 5 samples in Figures 4D and 4E. The quantification and statistical analysis for data shown in these figures need to be strengthened.

Reply: The spatial extent and density of the population, as well as the resolution of the acquired images, made it impossible to measure the activity of every neuron in the population. Individual cells were thus picked randomly from amongst the most visually accessible. No data were excluded from the analysis and representative examples were shown in the figures. In total we analysed between 50 and 59 random cells per brain for the control condition and 53 to 74 for *prosRNAi* brains, which is now also stated in the Methods (p. 29). However, for better clarity we only show 5 example neurons each in Figure 4E. We now show an additional figure with neuronal responses in a further pair of brains in EV5. There is biological variability in both control and *pros* knock-down brains, which we tried to convey in the figures by depicting a range of responses. We did not observe any non-responding cells in either control or mutant brains, but the referee is correct in saying that we cannot exclude that some neurons did not respond.

N~5 brains is customary in electrophysiology experiments (see for example Frank *et al.* (2017) *Curr Biol* 27:2381-8 and Enoki *et al.* (2017) *PNAS* 114:E2476-85; in flies and mice, respectively) and all the neurons in the brains consistently responded within a minute to picrotoxin. Figure 4D thus shows the average population response of 5 pairs of brains to the picrotoxin treatment demonstrating their similarity. We tested for normality using the KS test and since both columns passed ($p > 0.1$) used an unpaired two-tailed t-test: $p = 0.6068$ (Mann Whitney test also gives $p = 0.4206$).

We have also added to to Figure 4 legend that “solid line is mean, shaded area is S.E.M.” and “unpaired two-tailed t-test: $p = 0.6068$ ”.

Thank you for submitting your revised manuscript to The EMBO Journal. I have now heard back from the referees. As you can see below, the referees appreciate the introduced changes and I am therefore very happy to accept the manuscript for publication here. Before we can transfer the manuscript to our publisher there are a few things we should sort out. I have provided a link below so that you can upload the revised version.

REFeree REPORTS

Referee #1:

Overall, I am satisfied with the revisions the authors have made and their efforts to address the concerns of the reviewers experimentally and by appropriate textual adjustments. I have no further concerns to raise.

Referee #2:

The revised manuscript and point-by-point response have addressed my concerns. I support its publication in EMBO J.

Corresponding Author Name: Rita Sousa-Nunes

Journal Submitted to: The EMBO Journal

Manuscript Number: EMBOJ-2017-98163R